Morphology of the jaw, suspensorial, and opercle musculature of Beloniformes and related species (Teleostei: Acanthopterygii), with a special reference to the m. adductor mandibulae complex

Werneburg Ingmar i.werneburg@gmail.com
Museum für Naturkunde, Leibniz-Institut für Evolutions- & Biodiversitätsforschung an der Humboldt-Universität zu Berlin , Berlin , Germany
Wilson Laura
Electronic publication date: 2015 Feb 24
Publication date: 2015
Volume: 3
Electronic Location ID: e769
Received 2014 Dec 11; Accepted 2015 Jan 26
Copyright: © 2015 Werneburg
Copyright year: 2015
Copyright holder: Werneburg
License: This is an open access article distributed under the terms of the Creative Commons Attribution License, which permits unrestricted use, distribution, reproduction and adaptation in any medium and for any purpose provided that it is properly attributed. For attribution, the original author(s), title, publication source (PeerJ) and either DOI or URL of the article must be cited.
License URL: https://creativecommons.org/licenses/by/4.0/

Keywords: Atherinomorpha, Oryzias, Perca, Belone, Jaw elongation, Feeding

Funding: SNF Advanced Postdoc Mobility P300P3_158526 The author was funded by SNF Advanced Postdoc Mobility Grant P300P3_158526. The funders had no role in study design, data collection and analysis, decision to publish, or preparation of the manuscript.

==============================
The taxon Beloniformes represents a heterogeneous group of teleost fishes that show an extraordinary diversity of jaw morphology. I present new anatomical descriptions of the jaw musculature in six selected beloniforms and four closely related species. A reduction of the external jaw adductor (A1) and a changed morphology of the intramandibular musculature were found in many Beloniformes. This might be correlated with the progressively reduced mobility of the upper and lower jaw bones. The needlefishes and sauries, which are characterised by extremely elongated and stiffened jaws, show several derived characters, which in combination enable the capture of fish at high velocity. The ricefishes are characterised by several derived and many plesiomorphic characters that make broad scale comparisons difficult. Soft tissue characters are highly diverse among hemiramphids and flying fishes reflecting the uncertainty about their phylogenetic position and interrelationship. The morphological findings presented herein may help to interpret future phylogenetic analyses using cranial musculature in Beloniformes.

Introduction

The m. adductor mandibulae complex belongs to one of the most intensively studied soft tissues in vertebrates. It primarily moves the skeletal elements associated to the mandibular arch and is the main head and the most powerful feeding musculature. The m. adductor mandibulae complex is highly adapted to different feeding strategies among vertebrate clades and, as such, experienced a large amount of diversification. Its anatomy is informative for different phylogenetic levels and a mutual evolution with jaw and skull anatomy can be observed (e.g., Gosline, 1986; Diogo, 2008; Diogo & Abdala, 2010; Datovo & Vari, 2013).

Among teleost fishes, the jaw anatomy of Beloniformes, the needlefishes and their allies, is very diverse. As such, they received reasonable attention in osteological, phylogenetic as well as ontogenetic analyses Rosen & Parenti, 1981; Boughton, Collette & McCune, 1991; Lovejoy & De Araujo, 2000; Lovejoy, Iranpour & Collette, 2004. The taxon includes small, short-snouted and duckbilled ricefishes (Adrianichthyidae) (Parenti, 1987), which live in flooded Asian rice fields. Halfbeaks (hemiramphids), another group, are characterised by an elongated lower jaw. The flying fishes (Exocoetidae) have short snouts; whereas the sauries (Scomberesocidae) and needlefishes (Belonidae), which are adapted to fast swimming and fish hunting, have elongated upper and lower jaws with extended teeth rows (Nelson, 2006). Although the drastic ontogenetic changes of the jaws have been previously studied in their external shape (Boughton, Collette & McCune, 1991; Lovejoy, Iranpour & Collette, 2004), the anatomy of the fully formed cranial musculature has received little attention.

Beloniformes belong to the Atherinomorpha (Fig. 1), which are placed within the Acanthopterygii. The phylogenetic relationships among acanthopterygian groups, which also include taxa such as Perciformes and Mugilomorpha, are controversial (e.g., Stiassny, 1990; Johnson & Patterson, 1993; Parenti, 1993; Parenti & Grier, 2004; Rosen & Parenti, 1981; Wu & Shen, 2004; Nelson, 2006; Setiamarga et al., 2008; Near et al., 2013). Smegmamorpha, Mugilomorpha, or Paracanthopterygii have all been hypothesised to form the sister taxon to Atherinomorpha.

Figure 1 Alternative topologies for atherinomorph interrelationship as referred in the literature.

(A) Rosen (1964), (B) Rosen & Parenti (1981), (C) Lovejoy, Iranpour & Collette (2004). Note the different arrangement of Cyprinodontea (6), Hemiramphidae, and the position of Scomberesocidae; corresponding taxa are highlighted. Numbers of non-terminal clades: 1, Atherinomorpha (1*: clade named as “Atheriniformes” by Rosen, 1964); 2, Cyprinodontoidei; 3, Exocoetoidei; 4, Exocoetoidea; 5, Scomberesocoidea; 6, Cyprinodontea; 7, Beloniformes, 8, N.N.

The monophyly of Atherinomorpha is currently accepted (Nelson, 2006; Near et al., 2013). Atheriniformes form the sister group of Cyprinodontea, which comprises Cyprinodontiformes (killifishes and their allies) and Beloniformes (Figs. 1B–1C). Recently, Li (2001) analysed osteological data of the hyobranchial apparatus and re-established the traditional hypotheses of Berg (1958) and Rosen (1964) of a closer relationship of Adrianichthyidae to Cyprinodontiformes (Fig. 1A; see also Temminck & Schlegel, 1846: compared to Yamamoto, 1975). This hypothesis, however, was not based on a cladistic analysis and represents phenetic classifications. These classifications are in strong contrast to several morphological and molecular analyses, which result in a sister group relationship of Adrianichthyidae and Exocoetoidea, comprising the remaining Beloniformes (Fig. 1), and Beloniformes as the sister group of Cyprinodontiformes (Rosen & Parenti, 1981; Collette et al., 1984; White, Lavenberg & McGowen, 1984; Naruse et al., 1993; Dyer & Chernoff, 1996; Naruse, 1996; Hertwig, 2008).

The phylogenetic relationships within Beloniformes are still a matter of debate. Traditional studies (Rosen, 1964; Rosen & Parenti, 1981) found two major clades within Beloniformes (excl. Adrianichthyidae), namely Exocoetoidea (flying fishes and halfbeaks) and Scomberesocoidea (sauries and needlefishes), together forming the Exocoetoidei (Rosen, 1964; Parin & Astakhov, 1982; Collette et al., 1984; Figs. 1A–1B).

Recently, Lovejoy (2000) and Lovejoy, Iranpour & Collette (2004) proposed the paraphyly of hemiramphids and nested Scomberesocidae inside “Belonidae” (Fig. 1C). The paraphyly of hemiramphids was also supported by Tibbetts (1991) and Aschliman, Tibbetts & Collette (2005). The halfbeak Dermogenys (which is included in the present study) was found to be a member of the Zenarchopteridae, which comprise a subset of hemiramphids of the Indo-West-Pacific (Anderson & Collette, 1991; Lovejoy, 2000; Meisner, 2001). Zenarchopteridae represents the sister taxon of the clade formed by needlefishes and sauries (Lovejoy, Iranpour & Collette, 2004; Aschliman, Tibbetts & Collette, 2005). Other representatives of the traditionally recognized hemiramphids grouped with the Exocoetidae, or as the sister group to the clade Zenarchopteridae + “Belonidae” (Fig. 1C).

The complex jaw musculature of Beloniformes has only been studied in very few species so far, and most published descriptions of beloniform species are superficial and insufficiently illustrated, making broad scale phylogenetic comparisons impossible. That makes broad phylogenetic comparisons impossible. The aim of the present study was to illustrate and describe the morphological diversity of cranial musculature of six selected species of Beloniformes in great detail and to compare it to external jaw anatomy. By using manual dissections and histological slide sections I aim to provide a comprehensive anatomical basis for future researchers studying more species in a phylogenetic context.

In the present, purely anatomical study, the great diversity within beloniform subgroups or within non-beloniform groups could not be studied by maintaining the provided extent and detail of illustrations and descriptions. However, I present some considerations about the potential phylogenetic relevance of some characters that have to be tested in future studies. Therefore, four selected near related acanthopterygian species, which may serve as outgroup in future phylogenetic studies, are described. In addition to two atherinomorph species, I included the percomorph Perca fluviatilis, which was recently used to define the ancestral pattern of atherinomorph jaw musculature (Hertwig, 2008), and the mugilomorph Rhinomugil corsula, which is possibly closer related to Atherinomorpha (Stiassny, 1990; Setiamarga et al., 2008; Near et al., 2013). A preliminary character mapping is presented.

Materials and Techniques

Taxonomic sampling

The cranial anatomy of ten acanthopterygian species was studied, including six species of Beloniformes (Figs. 2–20). Specimens from the following collections were used: Phyletisches Museum der Friedrich Schiller Universität Jena, Germany (ISZE), Smithsonian Institution of the National Museum of Natural History Washington, USA (USNM), Naturhistorisches Museum der Burgergemeinde Bern, Switzerland (NMBE).

• Perciformes, Perca fluviatilis (Linnaeus, 1758) (ISZE) (Figs. 2A and 5);

• Mugilomorpha, Rhinomugil corsula (Hamilton, 1822) (NMBE) (Figs. 2B, 6 and 7);

• Atheriniformes, Atherina boyeri (Risso, 1810) (NMBE) (Figs. 2C, 8, 9 and 12D);

• Cyprinodontiformes, Aplocheilus lineatus (Valenciennes, 1846) (NMBE) (Figs. 2D, 10 and 11);

• Beloniformes, Adrianichthyidae, Oryzias latipes (Temminck & Schlegel, 1846) (NMBE) (Figs. 2E, 12A–12C, 13);

• Beloniformes, Adrianichthyidae, Xenopoecilus oophorus (Kottelat 1990) (NMBE) (Fig. 3);

• Beloniformes, Exocoetidae, Parexocoetus brachypterus (Richardson, 1846) (USNM 299385) (Figs. 2F, 14 and 15);

• Beloniformes, Hemiramphidae, Dermogenys pusilla (Kuhl and Van Hasselt, 1823) (NMBE) (Figs. 2G, 16 and 17);

• Beloniformes, Belonidae, Belone belone (Linnaeus, 1761) (NMBE) (Figs. 2H and 18);

• Beloniformes, Scomberesocidae, Scomberesox saurus (Walbaum, 1782) (ISZE) (Figs. 19 and 20).

Figure 2 Overview on the cranial anatomy in the eight species manually dissected in this study.

Skin is removed. Abbreviations of muscles (m., musculus) and selected bones: A1, external section of m. adductor mandibulae; A2/3, internal section of m. adductor mandibulae; AAP, m. adductor arcus palatini; den, dentary; DO, m. dilatator operculi; EA, epaxial musculature; lac, lacrimal; LAP, m. levator arcus palatini; LO, m. levator operculi; max, maxilla; op, opercle; PH, m. protractor hyoidei; pop, preopercle; T, m. trapezius. Drawings not to scale. For detailed labelling, scales, histological sections, and further illustrations see Figs. 5–20.

Figure 3 The duckbilled ricefish Xenopoecilus oophorus (Beloniformes, Adrianichthyidae).

Serial sections through the head. Slice thickness, 12 µm. Section numbers: (A) 14, (B) 170, (C) 206, (D) 268, (E) 340, (F) 440, (G) 450 (lenses redrawn), (H) 586, (I) 648, (J) 698. Bar scale provided for (A–J). Magnifications B’, D’, I’–J’ are not to scale. Compare to the other adrianichthyid studied herein, Oryzias latipes (Figs. 2E, 12A–12C and 13).

Figure 4 Phylogenetic framework.

Arrangement of the species studied herein and those from the literature (*) used for the reconstruction of character evolution (character mapping); following Lovejoy, Iranpour & Collette (2004; compare to Fig. 1C). Outlines indicate the species, which were manually dissected herein; not to scale (compare to Figs. 5–20).

Figure 5 Perca fluviatilis.

(A–D) Manual dissections. Compare to Fig. 2A.

Figure 6 Rhinomugil corsula.

(A–D) Manual dissections; levels of histological sections (Fig. 7) are indicated. Compare to Fig. 2B.

Figure 7 Rhinomugil corsula.

(A–D) Histological sections; compare to Fig. 6.

Figure 8 Atherina boyeri.

(A–D) Manual dissections; levels of histological sections (Fig. 9) are indicated. For one further manual dissection of this species see Fig. 12D. Compare to Fig. 2C.

Figure 9 Atherina boyeri.

(A–D) Histological sections; compare to Figs. 8 and 12D.

Figure 10 Aplocheilus lineatus.

(A–D) Manual dissections; levels of histological sections (Fig. 11) are indicated. Compare to Fig. 2D.

Figure 11 Aplocheilus lineatus.

(A–D) Histological sections; compare to Fig. 10.

Figure 12 Manual dissections.

(A–C) Oryzias latipes; levels of histological sections (Fig. 13) are indicated. (A) and (C) modified from Werneburg & Hertwig (2009). Compare to Fig. 2E. (D) Atherina boyeri; for other dissections of this species see Fig. 8, for histological sections see Fig. 9. Compare to Fig. 2C.

Figure 13 Oryzias latipes.

(A–D) Histological sections; compare to Fig. 12. Modified from Werneburg & Hertwig (2009).

Figure 14 Parexocoetus lineatus.

(A–D) Manual dissections; levels of histological sections (Fig. 15) are indicated. Compare to Fig. 2F.

Figure 15 Parexocoetus lineatus.

(A–D) Histological sections; compare to Fig. 14.

Figure 16 Dermogenys pussila.

(A–C) Manual dissections; levels of histological sections (Fig. 17) are indicated. Compare to Fig. 2G.

Figure 17 Dermogenys pussila.

(A–D) Histological sections; compare to Fig. 16.

Figure 18 Belone belone.

(A–C) Manual dissections. Compare to Fig. 2H.

Figure 19 Scomberesox saurus.

(A–D) Manual dissections in an adult specimen; approximate levels of histological sections of a juvenile (Fig. 20) are indicated.

Figure 20 Scomberesox saurus.

(A–D) Histological sections in a juvenile specimen; compare to Fig. 19

For a phylogenetic analysis, published information on further beloniform, namely hemiramphid species, Hyporhamphus unifasciatus (Ranzani, 1841), Nomorhamphus sp. aff. ravnaki (Brembach, 1991), and Hemirhamphodon phaiosoma (Bleeker, 1852), were included (Table 1, Fig. 4). According to the new findings of Werneburg & Hertwig (2009), the data on O. latipes were modified when compared to Hertwig (2005), Hertwig (2008) and Werneburg (2007).

Table 1 For the phylogenetic arrangement of species see Fig. 4.

Character complex	Character	Aplocheilus lineatus	Atherina boyeri	Belone belone	Dermogenys pusilla	Hemirhamphodon phaiosoma*	Hyporhamphus unifasciatus*	Nomorhamphus sp. aff. ravnaki*	Oryzias latipes	Parexocoetus brachypterus	Perca fluviatilis	Rhinomugil corsula	Scomberesox saurus	Xenopoecilus oophorus	
External section of m. adductor
mandibulae (A1)	General appearance	0	0	1	1	0	1	0	0	1	0	0	1	0	
Orientation	2	3	X	X	X	2	X	2	X	1	0	X	2	
Insertion	4	3	X	X	X	1	X	2	X	0	1	X	1	
Internal section of m. adductor
mandibulae (A2/3)	Origin	1	X	1	0	0	0	0	1	0	0	1	1	1	
Lateral head	0	X	0	1	0	0	0	3	2	2	0	0	3	
Medial head	2	X	4	4	X	X	X	3	2	0	1	4	3	
Intermedial head	1	X	1	X	X	X	X	2	X	X	0	1	2	
Muscle portions	0	1	0	0	0	0	0	0	0	0	0	0	0	
Orientation of muscle heads	0	1	0	0	?	?	?	2	2	0	2	0	0	
Relative size of muscle heads	2	X	3	0	?	?	?	3	1	0	2	3	3	
Insertion	0	0	1	0	0	0	0	0	0	0	0	1	0	
Intramandibular portion	2	3	0	0	0	0	0	0	0	4	1	0	0	
Intramandibular section of m. adductor
mandibulae (Aω)	Origin	3	3	2	0	0	0	0	1	0	4	2	2	5	
Shape	3	1	1	1	2	2	2	2	2	1	0	1	X	
Insertion	2	0	1	0	?	?	?	3	0	0	X	1	X	
M. intermandibularis	Cross section	2	0	2	2	2	2	2	1	2	1	1	2	1	
Shape	1	0	0	0	?	?	?	1	0	0	1	0	1	
M. protractor hyoidei	Origin	2	2	3	0	?	?	?	1	0	0	2	3	2	
Course	0	2	1	0	?	?	?	0	0	1	0	1	0	
Anterior part	0	1	1	1	?	?	?	0	1	1	2	1	0	
Insertion	0	1	2	2	2	2	2	2	0	2	2	0	2	
Insertion tendon	X	1	0	0	?	?	?	1	X	1	1	X	1	
M. adductor arcus palatini	Origin and insertion	0	1	1	0	1	1	1	0	1	1	0	1	0	
M. levator arcus palatini	Origin	4	3	1	0	?	?	?	2	0	4	1	1	2	
Course	1	1	1	1	?	?	?	0	1	1	0	1	0	
Relation to other muscles	0	X	2	1	?	?	?	0	0	0	0	2	0	
Insertion	0	2	0	0	?	?	?	1	0	0	2	0	1	
M. dilatator operculi	Origin	1	1	2	2	?	?	?	2	0	1	2	0	0	
Shape	0	0	1	1	?	?	?	0	1	1	0	1	1	
M. levator operculi	Origin	0	0	0	0	?	?	?	0	0	1	0	0	0	
Insertion	0	2	1	0	2	2	2	0	0	0	0	1	0	
Nerves	Truncus maxillaris infraorbitalis
trigemini	0	0	2	1	?	?	?	1	1	0	0	2	1	
Ramus mandibularis facialis	0	0	0	1	?	?	?	1	1	1	1	0	?	
Ligaments	Lig. premaxillo-maxilla	1	1	0	1	0	0	0	1	1	1	1	0	1	
Primordial ligament	1	0	1	1	1	1	1	1	1	0	1	1	1	
Upper jaw/palatine ligament	0	0	2	2	?	?	?	0	0	1	2	2	0	
Lig. parasphenoido-suspensorium	1	0	1	1	?	?	?	1	1	0	1	0	1	
Notes.

* indicating literature data from Hertwig (2008).

X not applicable

? unknown

Anatomical observations

Standard procedures for histology and manual dissection are those used by Werneburg (2007) and Werneburg & Hertwig (2009).

For dissection, two or more specimens per species were used. In the first step of dissection (summarised in Fig. 2) the lateral view of the skinned head including all muscles in their unaltered place, including the jaw adductor musculature, opercle-, and suspensoric-related musculature, was documented. In the second step, the external section of m. adductor mandibulae (A1) was mostly removed and the course of the internal section of m. adductor mandibulae (A2/3) was depicted. Further steps of dissection did allow inspection of the symplectic in lateral view with the A2/3 completely or partly removed. Finally, the medial view of the jaw apparatus was documented with a focus on the musculature medial of the lower jaw, namely the intramandibular section of m. adductor mandibulae (Aω), the anterior part of m. protractor hyoidei, and m. intermandibularis.

Serial sections were prepared for all species (slice thickness = 12 µm), except for Pe. fluviatilis and B. belone due to the size of these species. The positions of the sections are indicated in the dissection figures (Figs. 6, 8, 10, 12, 14, 16 and 19). For S. saurus, a juvenile specimen was used for histological sectioning (Fig. 20), whereas for manual dissections and character coding (as for all species), adult specimens were used (Fig. 20).

Nomenclature

Osteological nomenclature follows Weitzman (1962) and Weitzman (1974) with modifications as summarised by Hertwig (2008). Basic myological terminology is that of Werneburg (2011). Fish muscle nomenclature mainly corresponds to that of Winterbottom (1974). The homologisation of particular muscular portions follows Werneburg & Hertwig (2009). The nomenclature of the nervous system refers to Holje, Hildebrand & Fried (1986). For osteological and, if available, for myological comparisons, I relied on Osse (1969) for Perciformes; on Thomson (1954) for Mugilomorpha; on Kulkarni (1948), Rosen (1964), Karrer (1967), Hertwig (2005) and Hertwig (2008) for Adrianichthyidae and Cyprinodontiformes; on Clemen, Wanninger & Greven (1997), Greven, Wanninger & Clemen (1997), Meisner (2001), and Shakhovskoi (2002) for hemiramphids; on Khachaturov (1983) and Shakhovskoi (2004) for Exocoetidae; and on Chapman (1943) for Scomberesocidae.

Character evolution

Using PAUP* (Swofford, 2003), a character mapping was performed. Therefore, the topology of Lovejoy, Iranpour & Collette (2004) was used as template to arrange the phylogeny of the beloniform species studied herein and of three additional hemiramphid species (Fig. 4; cf. Fig. 1C). For the interrelationship of major acanthopterygian groups, the present study follows the findings of Stiassny (1990), Setiamarga et al. (2008), and Near et al. (2013). Therein, Percomorpha form the sister taxon to Ovalentaria. Consequently a polarisation of characters is given. The topology for the character mapping was drawn using the move branch function in Mesquite 2.01 (Maddison & Maddison, 2011).

Results and Discussion

Characters and character mapping

In total, 37 soft tissue characters are described and discussed below. The character matrix can be found in Table 1. The results of the character mapping are listed in Table 2. Therein, the consensus of Acctran and Deltran optimizations are documented. Due to the particular focus on the morphological descriptions and illustration of this study, the taxonomic sampling is limited. Also the available data from the literature record is limited. As such, I avoid discussing the character changes in detail. They should serve as summary of character distribution of the species studied herein. The phylogenetic relevance of the characters should be subject of evaluation and discussion in future, more quantitative analyses of the cranial musculature of Beloniformes. Those studies may also consider more closely related species for the comparison with Atherinomorpha.

Table 2 Character evolution within the topology of Lovejoy, Iranpour & Collette (2004) (Fig. 4).

Character complex	Character	Plesiomorphic state in taxon 1	→	Derived state in taxon 2	
		Ovalentaria	→	Rhinomugil corsula	
Internal section of m. adductormandibulae (A2/3)	Spatial orientation	The medial head of A2/3 is situated dorsally to the lateral head or is at least clearly visible in lateral view [state 0].	⇒	The lateral head is situated laterally to the medial head and can cover it completely [state 2].	
Internal section of m. adductormandibulae (A2/3)	The medial head	Originates from the hyomandibular, the metapterygoid, and the symplectic, as well as from processus lateralis hyomandibularis [state 0].	⟶	Originates from the hyomandibular and from the metapterygoid [state 1].	
Internal section ofm. adductor mandibulae (A2/3)	Intramandibularportion	Absent [state 0].	⟶	Present and has a narrow insertion on the medial face of processus coronoideus dentalis [state 1].	
Intramandibular section ofm. adductor mandibulae (Aω)	Shape	Double-feathered muscle, in which one of the resulting muscle parts may project to a far caudad direction [state 1].	⇒	The lateral head inserts broadly to the medial face of the dentary and cartilago Meckeli. The medial head inserts ventrally to the medial face of the dentary and anteriorly to the medial face of the anguloarticular [state 0].	
M. intermandibularis	Shape	Parallel fibred with no tendinous origin at the dentary [state 0].	⟶	Spindle-shaped with tendinous origin at the dentary [state 1].	
M. protractor hyoidei	Anterior part	As broad as high [state 1].	⇒	The dorsal head is flat and the ventral head is as high as broad [state 2].	
M. levator arcus palatini	Origin	From the autosphenotic and with some fibres at the sphenotic [state 4]	⇒	On a ridge of the sphenotic, the processus sphenoticus, and some fibres originate directly on the sphenotic [state 1]	
M. levator arcus palatini	Course	From origin to insertion, the thickness broadens more than twice [state 1].	⇒	Thickness hardly changes [state 0].	
M. levator arcus palatini	Insertion	On the lateral face of the suspensoric to the hyomandibular and to the metapterygoid and with some fibres, it also can attach anteriorly to the processus lateralis hyomandibularis [state 0].	⟶	On the hyomandibular, anteriorly to the processus lateralis hyomandibularis, to the metapterygoid, and to the broad face of the preopercular [state 2].	
M. dilatator operculi	Origin	Laterally at the sphenotic, at the autosphenotic, and with some fibres possibly at the anteroventral area of the pterotic [state 1].	⇒	Laterally at the sphenotic and anteriorly at the lateral face of the pterotic [state 2].	
Ligaments	Primordial ligament	Present as a lig. maxillo-anguloarticulare between the maxilla and the anguloarticular [state 0].	⟶	Absent [state 1].	
Ligaments	Upper jaw/palatineligament	Present as lig. palato-maxilla between palatine and maxilla [state 0].	⟶	Absent [state 2]	
Ligaments	Lig. parasphenoido-suspensorium	Present [state 0].	⟶	Absent [state 1]	
		Ovalentaria	→	Atherinomorpha	
Intramandibular section ofm. adductormandibulae (Aω)	Origin	Broadly on the medial face of the quadrate and a part of the muscle can have a tendinous origin [state 2].	⟶	With a tendon anteroventrally to the medial face of the quadrate [state 3].	
		Atherinomorpha	→	Atherina boyeri	
External section of m. adductormandibulae (A1)	Insertion	To the medial face of the middle region of the maxilla [state 1].	⇒	With three tendons on the processus primordialis (anguloarticularis), to the medial side of the lacrimal, and medially to the anterodorsal tip of the maxilla [state 3].	
Internal section of m. adductormandibulae (A2/3)	Muscle portions	Does not separate in two portions [state 0].	⇒	Laterally separated into two portions [state 1].	
M. protractor hyoidei	Insertion	Dorsally as well as ventrally of m. intermandibularis to the dentary [state 2].	⇒	Ventrally to m. intermandibularis at the dentary [state 1].	
M. protractor hyoidei	Course	A fusion with the contralateral m. protractor hyoidei occurs at the level of the jaws or suspensoric and united, they travel rostrad and anteroventrally at the fused mm. protractor hyoidei a tendon can be formed on each side [state 0].	⇒	At the level of the anguloarticular, the muscles fuse only in their ventral regions; they separate on the level of the dentary in order to insert independently of the contralateral muscle to the dentary [state 2].	
M. adductor arcus palatini	Origin and insertion	The anterior portion originates along the whole parasphenoid and inserts dorsally along the entire suspensoric (in addition to other small attachments) [state 0].	⟶	Originates on the posterior part of the parasphenoid and inserts on the posterior region of the suspensoric [state 1].	
M. levator arcus palatini	Origin	From the autosphenotic and with some fibres at the sphenotic [state 4].	⇒	Ventrally at the dermosphenotic [state 3].	
M. levator arcus palatini	Insertion	On the lateral face of the suspensoric on the hyomandibular and to the metapterygoid and with some fibres, it also can attach anteriorly to the processus lateralis hyomandibularis [state 0].	⟶	On the hyomandibular, anteriorly to the processus lateralis hyomandibularis, to the metapterygoid, and to the broad face of the preopercular [state 2].	
M. levator operculi	Insertion	Dorsally to the medial face of the opercle with a continuous horizontal level of insertion [state 0].	⇒	Dorsally to the medial face and dorsally to the lateral face of the opercle [state 2].	
		Atherinomorpha	→	Cyrinodontea	
Ligaments	Primordial ligament	Present as a lig. maxillo-anguloarticulare between the maxilla and the anguloarticular [state 0].	⟶	Absent [state 1].	
Ligaments	Lig. parasphenoido-suspensorium	Present [state 0].	⟶	Absent [state 1].	
		Cyrinodontea	→	Aplocheilus lineatus	
External section of m. adductormandibulae (A1)	Insertion	On the medial face of the middle region of the maxilla [state 1].	⇒	With two tendons to the lateral face of the medial part of the maxilla and to the medial face of the lacrimal [state 4].	
Internal section of m. adductormandibulae (A2/3)	Intramandibular portion	Absent [state 0].	⟶	Present with broad insertions to the processus coronoideus dentalis, to cartilago Meckeli, and to the anguloarticular [state 2].	
Intramandibular section ofm. adductormandibulae (Aω)	Insertion	On the medial face of the lower jaw, the Aω (when not differentiated into heads) inserts broadly to the dentary, cartilago Meckeli and/or to the anguloarticular [state 0]v	⇒	On the ventral part of the dentary [state 2].	
M. protractor hyoidei	Insertion	Dorsally as well as ventrally of m. intermandibularis to the dentary [state 2].	⇒	Dorsally to the insertion of m. intermandibularis at the dentary and covers at least the posterodorsal area of the latter muscle [state 0].	
		Cyrinodontea	→	Beloniformes	
M. dilatator operculi	Origin	Laterally at the sphenotic, at the autosphenotic, and with some fibres possibly at the anteroventral area of the pterotic [state 1].	⇒	Ventrally at the lateral face of the sphenotic [state 0].	
Nerves	Truncus maxillaris infraorbitalis trigemini	Branches into the ramus mandibularis trigemini and ramus maxillaris trigemini short before or after leaving the neurocranium [state 0].	⇒	First branches at the level of the eye [state 1].	
		Beloniformes	→	Adrianichthyidae	
Internal section of m. adductormandibulae (A2/3)	Lateral head	Originates almost overall at the vertical aspect of preopercle, at the posterior part of the horizontal aspect of the preopercle, as well as on the processus lateralis hyomandibularis [state 0].	⇒	With a narrow attachment, it only originates on the ventral third of the vertical aspect of the preopercle [state 3].	
Internal section of m. adductormandibulae (A2/3)	Medial head	Originates only from the metapterygoid [state 2].	⇒	Arises from the lateral faces of the quadrate, the symplectic, and the cartilaginous interspaces of the hyopalatine arch, and from the tendon of the m. adductor arcus palatini quadrati [state 3].	
Internal section of m. adductormandibulae (A2/3)	Intermedial head	Originates from the horizontal aspect of the preopercle and at the processus caudalis quadrati [state 1].	⇒	Originates only on the processus caudalis quadrati [state 2].	
M. levator arcus palatini	Course	During its course from origin to insertion, the thickness broadens more than twice [state 1].	⇒	Thickness hardly changes [state 0].	
M. levator arcus palatini	Insertion	On the lateral face of the suspensoric, to the hyomandibular and to the metapterygoid and with some fibres, it also can attach anteriorly to the processus lateralis hyomandibularis [state 0].	⇒	On the broad face of praeopercular and posterodorsally to the symplectic [state 1].	
		Adrianichthyidae	→	Oryzias latipes	
External section of m. adductormandibulae (A1)	Insertion	To the medial face of the middle region of the maxilla [state 1].	⇒	To the posterior edge of the dentary [state 2].	
5 Internal section of m. adductormandibulae (A2/3)	Spatial orientation	The medial head of A2/3 is situated dorsally to the lateral head or is at least clearly visible in lateral view [state 0].	⇒	The lateral head is situated laterally to the medial head and can cover it completely [state 2].	
M. protractor hyoidei	Origin	Ventrally to the ceratohyal [state 2].	⇒	With two heads ventrally and laterally at the ceratohyal and at the anterior tips of the branchiostegal rays [state 1].	
M. dilatator operculi	Origin	Ventrally at the lateral face of the sphenotic [state 0].	⟶	Laterally at the sphenotic and anteriorly at the lateral face of the pterotic [state 2].	
		Adrianichthyidae	→	Xenopoecilus oophorus	
		-		-	
		Beloniformes	→	Exocoetoidea	
External section of m. adductormandibulae (A1)	General appearance	Present [state 0].	⇒	Absent [state 1].	
Internal section of m. adductormandibulae (A2/3)	Origin	With three muscle heads in its origin (A2/3, lateral; A2/3, medial; A2/3, intermedial) [state 1].	⇒	With two muscle heads (A2/3, lateral; A2/3, medial) in its origin [state 0].	
M. protractor hyoidei	Origin	Medially to the ceratohyal [state 2].	⇒	Laterally at the ceratohyal [state 0].	
M. adductor arcus palatini	Origin and insertion	Its anterior portion originates along the whole parasphenoid and inserts dorsally along the entire suspensoric [state 0].	⟶	Its anterior portion originates on the posterior part of the parasphenoid and inserts on the posterior region of the suspensoric [state 1].	
		Exocoetoidea	→	Parexocoetus lineatus	
5 Internal section of m. adductormandibulae (A2/3)	Orientation of muscle heads	The medial head is situated dorsally to the lateral head or is at least clearly visible in lateral view [state 0].	⇒	The lateral head is situated laterally to the medial head and can cover it completely [state 2].	
Internal section of m. adductormandibulae (A2/3)	Relative size of muscle heads	The medial head is larger than the lateral head [state 3].	⇒	The medial head is relatively narrow when compared to the lateral head [state 1].	
Internal section of m. adductormandibulae (A2/3)	Lateral head	Originates almost overall at the vertical aspect of preopercle, at the posterior part of the horizontal aspect of the preopercle, as well as on the processus lateralis hyomandibularis [state 0].	⇒	Originates ventrally at the processus lateralis hyomandibularis, at the ventral third of the vertical aspect of the preopercle, as well as on the processus caudalis quadrati [state 2].	
M. protractor hyoidei	Insertion	Dorsally as well as ventrally of m. intermandibularis to the dentary [state 2].	⇒	Dorsally to the insertion of m. intermandibularis at the dentary and covers at least the posterodorsal area of the latter muscle [state 0].	
		Exocoetoidea	→	clade A	
M. levator operculi	Insertion	Dorsally to the medial face of the opercle and has a continuous horizontal level of insertion [state 0].	⇒	Dorsally to the medial face and dorsally to the lateral face of the opercle [state 2].	
Ligaments	Lig. premaxillo-maxilla	Spans between the proximal ends of the premaxilla and the maxilla [state 1].	⇒	Spans broadly between premaxilla and maxilla [state 0].	
		clade A	→	clade B	
		-		-	
		clade B	→	Zenarchopteridae	
		-		-	
		clade C	→	Demogenys pussila	
Internal section of m. adductormandibulae (A2/3)	Lateral head	Originates almost overall at the vertical aspect of preopercle, at the posterior part of the horizontal aspect of the preopercle, as well as on the processus lateralis hyomandibularis [state 0].	⇒	Originates at the vertical aspect of the preopercle (but does not reach its dorsal most tip) and at more than half of the horizontal aspect of the preopercle [state 1].	
Intramandibular section of m. adductormandibulae (Aω)	Shape	A parallel fibred muscle [state 2].	⇒	A double-feathered muscle[state 1].	
M. adductor arcus palatini	Origin and insertion	Its anterior portion originates on the posterior part of the parasphenoid and inserts on the posterior region of the suspensoric [state 1].	⇒	Its anterior portion originates along the whole parasphenoid and inserts dorsally along the entire suspensoric [state 0].	
M. levator operculi	Insertion	Dorsally to the medial face and dorsally to the lateral face of the opercle [state 2].	⇒	Dorsally to the medial face of the opercle and has a continuous horizontal level of insertion[state 0].	
Ligaments	Lig. premaxillo-maxilla	Spans broadly between premaxilla and maxilla [state 0].	⇒	Spans between the proximal ends of the premaxilla and the maxilla [state 1].	
		clade C	→	Nomorhamphus sp.	
		-		-	
		Zenarchopteridae	→	Hemirhamphodon phaisoma	
		-		-	
		clade B	→	clade D	
Internal section of m. adductormandibulae (A2/3)	Origin	Two muscle heads (A2/3, lateral; A2/3, medial) in its origin [state 0].	⇒	Three muscle heads (A2/3, lateral; A2/3, medial; A2/3, intermedial) in its origin [state 1]	
Internal section of m. adductormandibulae (A2/3)	Insertion	Only on the medial side of the lower jaw [state 0].	⇒	Also on the coronomeckelian bone [state 1].	
Intramandibular section of m. adductormandibulae (Aω)	Origin	With a tendon anteriorly at the medial face of the symplectic [state 0].	⇒	Broadly on the medial face of the quadrate and a part of the muscle can have a tendinous origin[state 2].	
Intramandibular section of m. adductormandibulae (Aω)	Shape	A parallel fibred muscle [state 2].	⇒	A double-feathered muscle, in which one of the muscle parts may project to a far caudad direction [state 1].	
Intramandibular section of m. adductormandibulae (Aω)	Insertion	If not differentiated into heads, on the medial face of the lower jaw, broadly to the dentary, cartilago Meckeli and/or to the anguloarticular [state 0].	⇒	Broadly to the dentary, to the anguloarticular, and to the cartilago Meckeli; a ventral part in feathered muscles inserts far anteriorly to the medial face of the dentary [state 1].	
M. protractor hyoidei	Origin	Laterally at the ceratohyal [state 0].	⇒	Medially to the ceratohyal [state 3].	
M. protractor hyoidei	Course	A fusion with the contralateral m. protractor hyoidei occurs at the level of the jaws or suspensoric and united, they travel rostrad and anteroventrally at the fused mm. protractor hyoidei a tendon can be formed on each side [state 0]	⇒	Such a fusion does not occur[state 1].	
M. levator arcus palatini	Origin	Broadly on the sphenotic [state 0].	⇒	On a ridge of the sphenotic, the processus sphenoticus, and some fibres originate directly on the sphenotic [state 1].	
M. levator operculi	Insertion	Dorsally to the medial face and dorsally to the lateral face of the opercle [state 2].	⇒	Also dorsally at the medial face of the opercle, but it attaches more ventrally to the anterior region of the medial face of the opercle[state 1]	
Nerves	Truncus maxillaris infraorbitalis trigemini	First branches at the level of the eye into the ramus mandibularis trigemini and ramus maxillaris trigemini [state 1].	⇒	Branches already within the neurocranium. Afterwards, the ramus maxillaris trigemini splits into two branches. Dorsally to the posterior part of the suspensoric, the branches align laterally and medially along the course of ramus mandibularis trigemini. On the level of the jaw joint, the branches of ramus maxillaris trigemini change their course into an anterodorsad direction and enter the upper jaw. Ramus mandibularis trigemini travels anteroventrad to the lower jaw [state 2].	
Nerves	Ramus mandibularis facialis	Branches differently to state 0 [state 1]	⇒	Branches after leaving the hyomandibular laterally to the suspensoric in order to run with two branches to the medial side of the suspensoric [state 0]	
		clade D	→	Belone belone	
		-		-	
		clade D	→	Scomberesox saurus	
M. protractor hyoidei	Insertion	Dorsally as well as ventrally of m. intermandibularis to the dentary [state 2].	⇒	Dorsally to the insertion of m. intermandibularis at the dentary and covers at least the posterodorsal area of the latter muscle [state 0].	
Ligaments	Lig. parasphenoido-suspensorium	Absent [state 1].	⇒	Present [state 0].	
Notes.

→ direction of character change from taxon 1 to taxon 2

⇒ unambiguous character change

⟶ ambiguous character change

External section of the m. adductor mandibulae complex (A1)

The m. adductor mandibulae is differentiated into different muscle sections in teleost fishes, representing a complex of individual muscles, each having a separated origin, course, and insertion (Diogo, 2008; Diogo & Abdala, 2010). The external section of m. adductor mandibulae complex, A1, is the lateral-most jaw muscle. If present, it originates posteriorly on the suspensorium and/or on the preopercle, it runs rostrad, and has a tendinous insertion to the upper or lower jaw (i.e., Allis, 1897).

General appearance. An A1 is present in Perca fluviatilis (Figs. 2A and 5), Rhinomugil corsula (Figs. 2B, 6 and 7), Atherina boyeri (Figs. 2C, 8, 9 and 12D), Aplocheilus lineatus (Figs. 2D, 10 and 11), Oryzias latipes (Figs. 2E, 12A–12C and 13), and Xenopoecilus oophorus (Fig. 3) [character state 0] but is absent in all other species studied herein, namely Dermogenys pussila, Parexocoetus brachypterus, Belone belone, and Scomberesox saurus [state 1].

In O. latipes, Hertwig (2008) and Werneburg & Hertwig (2009) described a lateral muscle of the adductor complex with an insertion to the lower jaw. It could be interpreted in two different ways: First, it could represent A1, the possession of which is plesiomorphic; A1 is present in all non-beloniform fishes studied and in O. latipes, it autapomorphically would have shifted its insertion to the lower jaw. Second, A1 could be reduced in O. latipes (Hertwig, 2005). In that case, one additional step of transformation would be needed, as the internal section of m. adductor mandibulae (A2/3) would be modified secondarily. Hertwig (2005) followed the principle of parsimony and opted for the first explanation. Werneburg (2007) interpreted an insertion of A1 to the maxilla and homologised the muscle to the A1 of the outgroup representatives. After reanalysing, this finding was revised and A1 actually inserts on the posterior edge of the dentary at two-thirds of its height below the coronoid process of this bone and has contact via connective tissue to the lig. maxillo-mandibulare in this species (Werneburg & Hertwig, 2009). Previously, the latter connection was misinterpreted as an upper jaw insertion (Werneburg, 2007).

Wu & Shen (2004) mentioned a small ventrolateral portion of A1, their A1-VL, in two flying fish species. As Hertwig (2005: 39) already pointed out, the homologisations of those authors remain unclear. Moreover, the illustration of that portion is lacking. It appears that Wu & Shen (2004) may have confused this portion with the lateral subdivision of A2/3. Hertwig (2008, 149) wrote: ‘In an extensive comparative study of the m. adductor mandibulae in teleostean fishes, [the authors], however, did not mention a subdivision of A2/3 either in the Mugilomorpha or in the Atherinomorpha, but this is probably down to their limited taxon sample, which comprised only three species of the latter.’ If Wu & Shen (2004) actually identified the remainder of A1 as their A1-VL (supported by the fact that an insertion of A1-VL to the maxilla is present), a high interspecific variability may be hypothesised for the flying fishes.

Starks (1916) dissected a belonid species, Tylosurus acus, in which he described an A1-muscle. Following the present homologisation, however, that muscle clearly represents the lateral head of the muscle A2/3, which has a similar anatomy as found in B. belone (see also below) and S. saurus (Figs. 2H and 18–20).

Orientation. The spatial orientation of A1 to the more medial, internal section of m. adductor mandibulae (A2/3) is different among species. In R. corsula (Figs. 2B, 6 and 7), the A1 is situated ventrolaterally to the lateral head of A2/3 and three-fourths of this head are still visible in lateral view [state 0]. In Pe. fluviatilis, the muscle is situated dorsolateral to the internal section and the complete lateral head (A2/3, lateral) is not covered in lateral view (Figs. 2A and 5) [state 1]. A1 is situated completely lateral to the intermedial head of the internal section of m. adductor mandibulae (A2/3, intermedial) in Ap. lineatus (Figs. 2D, 10 and 11), O. latipes (Figs. 2E, 12A–12C and 13; see also Werneburg & Hertwig, 2009), and X. oophorus (Fig. 3) and the lateral head (A2/3, lateral) is only covered in its anterior region [2(2)]. Laterally in At. boyeri (Figs. 2C, 8, 9 and 12D), the A1 completely covers the internal section of m. adductor mandibulae (A2/3) [state 3].

For the ground pattern of Atherinomorpha, Hertwig (2005) proposed that the external (A1) and internal (A2/3) sections are situated next to each other in a horizontal plane. As an outgroup of Atherinomorpha, the author used Pe. fluviatilis, in which the A2/3-portions are situated above each other in a horizontal plane (Figs. 2A and 5). In the present study, R. corsula was dissected as an additional, potential outgroup species, which is closely related to Atherinomorpha. Similar to Atherinomorpha (sensu Hertwig, 2008), the A1 of that species also has to be interpreted to be lateral to the A2/3 in a horizontal plane. As such, that character has to be withdrawn as an autapomorphy of Atherinomorpha. More detailed observation among Percomorpha could identify the orientation of A1 to A2/3 in Pe. fluviatilis (Figs. 2A and 5) as autapomorphy of Percomorpha or only of that species. In the latter case, the ‘A1 in horizontal plane to A2/3’ would need to be interpreted as plesiomorphic among Acanthopterygii. Observations among Mugilomorpha could identify the orientation of A1–A2/3 as a homoplastic character of R. corsula and Atherinomorpha. If all members of Mugilomorpha had an A1 lateral to A2/3, and when following the phylogenetic hypothesis of Stiassny (1990), that spatial orientation would need to be interpreted as a synapomorphy of Mugilomorpha + Atherinomorpha.

Insertion. The tendon of A1 inserts on the lateral face of the anterior part of the maxilla in Pe. fluviatilis (Figs. 2A and 5) [state 0], to the medial face of the middle region of the maxilla in R. corsula (Figs. 2B, 6 and 7) and X. oophorus (Fig. 3) [state 1], and to the posterior edge of the dentary in O. latipes (Figs. 2E, 12A–12C and 13) [state 2]. With three tendons, A1 inserts on the processus primordialis (anguloarticularis), to the medial side of the lacrimal, and medially to the anterodorsal tip of the maxilla in At. boyeri (Figs. 2C, 8, 9 and 12D) [state 3]. The A1 inserts with two tendons to the lateral face of the medial part of the maxilla and to the medial face of the lacrimal in Ap. lineatus (Figs. 2D, 10 and 11) [state 4].

The insertion of A1 to the jaws is different in all species studied. A definition of homology (e.g., A1 inserts laterally to the maxilla) was not made, because the differences of A1 were too large. Hertwig (2008) observed several atherinomorph species and defined the insertion of A1 at the lateral face of the maxilla to be present in Pe. fluviatilis and “Aplocheilidae”. In contrast to Pe. fluviatilis (Figs. 2A and 5), however, the A1 inserts on the other end of the maxilla in Ap. lineatus (Figs. 2D, 10 and 11). The latter species has an additional tendon to the medial face of the lacrimal, a character which was found by Hertwig (2008) to be present in the ground pattern of Atherinomorpha (compare to Alexander, 1967; Parenti, 1993; Stiassny, 1990). For Cyprinodontiformes (incl. Aplocheilus), Hertwig (2005) was not able to define an unambiguous constellation of the insertion of A1. However, he argued that the insertion of A1 shifted based on the rotation of the maxilla in this taxon. As such, the insertion of A1 to the lateral face of the maxilla could be interpreted as being plesiomorphic among Atherinomorpha.

Internal section of the m. adductor mandibulae complex (A2/3)

The A2/3 usually originates with two or three muscle heads on the suspensoric and on the preopercle and inserts as a consistent muscle to the lower jaw. Muscle heads are defined as partial differentiations of a muscle. They have separated origins or insertions (Werneburg, 2007; Werneburg, 2011). Muscle heads gain a descriptive nomenclature herein; their position of origin (or insertion) and the spatial orientation were considered. This nomenclature differs from Winterbottom (1974), because that one is not applicable for muscle heads herein.

A2/3 can have an intramandibular portion. A muscle portion is defined as having a separate origin, course, and insertion, but as having some intertwining fibres or a shared tendon with another muscle portion of the same ontogenetic and/or phylogenetic origin (Werneburg, 2007; Werneburg, 2011).

Origin. In Pe. fluviatilis (Figs. 2A and 5), Pa. brachypterus (Figs. 2F, 14 and 15), and D. pussila (Figs. 2G, 16 and 17), the A2/3 has two muscle heads (A2/3, lateral; A2/3, medial) in its origin [state 0]. A2/3 originates with three muscle heads (A2/3, lateral; A2/3, medial; A2/3, intermedial) in R. corsula (Figs. 2B, 6 and 7), Ap. lineatus (Figs. 2D, 10 and 11), O. latipes (Figs. 2E, 12A–12C and 13), X. oophorus (Fig. 3), B. belone (Figs. 2H and 18), and S. saurus (Figs. 19 and 20) [state 1].

The cyprinodontiform species Ap. lineatus (Figs. 10 and 11) was found to have three muscle heads at its origin. This corresponds to the findings of Hertwig (2008). To confirm his findings, Hertwig (2008) used histological sections, which permit a much higher accuracy when distinguishing between minute muscle heads. I have seen many of the sections and used some herein, and can confirm his observations.

Jourdain (1878) described a specimen of B. belone (“vulgaris”), in which A2/3 was not separated. I dissected several specimens of that species and always found a separation, although I have to note that the differentiation of the lateral and the medial head were difficult. Also, apparently, Jourdain (1878) did not remove the lateral head of A2/3 as he expected A2/3 to represent an undifferentiated muscle mass and hence did not discover the intermedial head of A2/3.

The lateral head. The lateral head of A2/3 originates almost overall at the vertical aspect of preopercle, at the posterior part of the horizontal aspect of the preopercle, as well as on the processus lateralis hyomandibularis in R. corsula (Figs. 2B, 6 and 7), Ap. lineatus (Figs. 2D, 10 and 11), B. belone (Figs. 2H and 18), and S. saurus (Figs. 19 and 20) [state 0]. It originates at the vertical aspect of the preopercle (but does not reach its dorsal most tip) and at more than half of the horizontal aspect of the preopercle in D. pussila (Figs. 2G, 16 and 17) [state 1]. In Pe. fluviatilis (Figs. 3A and 5) and Pa. brachypterus (Figs. 2F, 14 and 15), the lateral head originates ventrally at the processus lateralis hyomandibularis, at the ventral third of the vertical aspect of the preopercle, as well as on the processus caudalis quadrati [state 2]. With a narrow attachment, it only originates on the ventral third of the vertical aspect of the preopercle in O. latipes (Figs. 2E, 12A–12C and 13) and X. oophorus (Fig. 3) [state 3].

Medial head. In Pe. fluviatilis, the medial head of A2/3 originates from the hyomandibular, the metapterygoid, and the symplectic, as well as from processus lateralis hyomandibularis (Fig. 5) [state 0]. It originates from the hyomandibular and from the metapterygoid in R. corsula (Figs. 6 and 7) [state 1] or only from the metapterygoid in Ap. lineatus (Figs. 10 and 11) and Pa. brachypterus (Figs. 2F, 14 and 15) [state 2]. It arises from the lateral faces of the quadrate, the symplectic, and the cartilaginous interspaces of the hyopalatine arch, and from the tendon of the m. adductor arcus palatini quadrati in O. latipes (Figs. 12A–12C and 13) and X. oophorus (Fig. 3) [state 3]. The medial head of A2/3 originates ventrally at the sphenotic, laterally at the hyomandibular, and dorsally at the metapterygoid in D. pussila (Figs. 16 and 17), B. belone (Figs. 2H and 18), and S. saurus (Figs. 19 and 20) [state 4].

Similar to the present study, Hertwig (2005) and Hertwig (2008) found the origin of the medial head of A2/3 to be highly variable. In addition to an adult specimen of S. saurus, a juvenile was studied (Figs. 9E–9H). In this specimen, a different orientation of the A2/3-heads was found (Werneburg, 2007). One could hypothesise that the medial head of A2/3 in the juvenile shifts its origin to a dorsal position and the intermedial head of A2/3 could shift its origin to a more ventral position (two transformation steps). Alternatively, the origin of the medial A2/3-head of the juvenile could shift ventrolaterally to the intermedial head of A2/3 and would be homologous to the intermedial head of A2/3 in the adult. Hence, the intermedial head of A2/3 in the juvenile (then the medial head of the adult) would keep its origin at the sphenotic (one transformation step). Those scenarios are very speculative because they are derived from only one observation. No final answer can be presented, because the variability of that character within S. saurus cannot be estimated. The species D. pussila, B. belone, and S. saurus show a very drastic ontogenetic elongation of the lower jaw (Hemiramphidae) or of both jaws (Belonidae, Scomberesocidae) (Boughton, Collette & McCune, 1991; Lovejoy, 2000; Lovejoy, Iranpour & Collette, 2004). It would be valuable to study if, correlated to the elongation of jaws, changes in the anatomy of the jaw musculature occur (origin, volume, course, insertion). Comparative ontogenetic and electromyographic studies (Focant, Jacob & Huriaux, 1981; Osse, 1969) could help to interpret the specific case mentioned herein. Ontogenetic changes in the anatomy of the jaw musculature were already observed by Hertwig (2005) in representatives of Goodeidae (Cyprinodontiformes: Crenichthys). Nanichthys (Scomberesocidae) is often not accepted as a ‘genus’ in a taxonomic sense and is often referred to as a dwarf morphotype of Scomberesox (Collette, 2004; Collette et al., 1984). However, if the juvenile specimen of S. saurus studied herein would actually represent a member of a valid genus Nanichthys, the arrangement of the A2/3-musculature may serve as a criterion to distinguish both species taxonomically.

Intermedial head. The intermedial head of A2/3 is situated between the lateral and the medial head. It originates only on the horizontal aspect of the preopercular in R. corsula (Figs. 6 and 7) [state 0]. It takes its origin from the horizontal aspect of the preopercle and at the processus caudalis quadrati in Ap. lineatus (Figs. 10 and 11), B. belone (Fig. 18), and S. saurus (Figs. 19 and 20) [state 1] and originates only on the processus caudalis quadrati in O. latipes (Figs. 12A–12C and 13) and X. oophorus (Fig. 3) [state 2]. An intermedial head is not present in Pe. fluviatilis, At. boyeri, Pa. brachypterus, and D. pussila.

Muscle portions. Unlike in all other species [state 0], A2/3 is laterally separated into two portions (by definition; see above and Werneburg, 2011) in At. boyeri (Figs. 8, 9 and 12D) [state 1]. The muscle portions of A2/3 have separated origins lateral at the posterior part of the suspensoric as well as separated insertions medial to the lower jaw. The medial portion of A2/3 is differentiated into two heads at its origin. The lateral portion of its A2/3 is not separated into heads. Among the species studied herein, and indeed, considering data from Hertwig (2008) regarding several other atherinid species, this condition has to be declared autapomorphic for At. boyeri (Atheriniformes).

Orientation of muscle heads. The spatial orientations of the medial and the lateral head of A2/3 are different among species. In Pe. fluviatilis (Figs. 2A and 5), Ap. lineatus (Figs. 10 and 11), D. pussila (Figs. 2G, 16 and 17), X. oophorus (Fig. 3), B. belone (Fig. 18), and S. saurus (Figs. 19 and 20), the medial head of A2/3 is situated dorsally to the lateral head or is at least clearly visible in lateral view [state 0]. The medial head of A2/3 is situated ventrally to the lateral head in At. boyeri (Figs. 8, 9 and 12D) [state 1]. The lateral head is situated laterally to the medial head and can cover it completely in R. corsula (Figs. 6 and 7), O. latipes (Figs. 12A–12C and 13), and Pa. brachypterus (Figs. 2F, 14 and 15) [state 2].

Relative size of muscle heads. The medial and the lateral heads of A2/3 have about the same size in Pe. fluviatilis (Fig. 5) and D. pussila (Figs. 16 and 17) [state 0]. The medial head is relatively narrow when compared to the lateral head in Pa. brachypterus (Figs. 14 and 15) [state 1]. The lateral head is quite widespread when compared to the medial head in R. corsula (Figs. 6 and 7) and Ap. lineatus (Figs. 10 and 11) [state 2]. The medial head is larger than the lateral head in O. latipes (Figs. 2E, 12A–12C and 13), X. oophorus (Fig. 3), B. belone (Figs. 2H and 18), and S. saurus (Figs. 19 and 20) [state 3].

Insertion. Except for B. belone (Figs. 2H and 18) and S. saurus (Figs. 19 and 20), A2/3 only inserts on the medial side of the lower jaw [state 0]. In the former species, it also inserts on the coronomeckelian bone [state 1], which is only found in these two species. It represents a bone, which is posterodorsally fused with the border of processus primordialis anguloarticularis. Both bones are separated from each other by a clear suture (Werneburg, 2007).

Intramandibular portion. An intramandibular portion of A2/3 is lacking in all Beloniformes [state 0]. It is present in R. corsula (Figs. 6 and 7) and has a narrow insertion on the medial face of processus coronoideus dentalis [state 1]. In Ap. lineatus (Figs. 10 and 11), it has broad insertions to the processus coronoideus dentalis, to cartilago Meckeli, and to the anguloarticular [state 2]. It inserts medially to the dentary in At. boyeri (Figs. 8, 9 and 12D) [state 3] and has a narrow insertion medially to the anguloarticular in Pe. fluviatilis (Fig. 5) [state 4].

The configuration of the intramandibular portion of A2/3 is different among non-beloniforms species studied here. As the criterion of homology, the intramandibular portion is defined to originate from an A2/3-associated aponeurosis or tendon herein. Hertwig (2008), who observed few species of Beloniformes (O. latipes and some hemiramphids), argued for an autapomorphic reduction of an intramandibular portion of A2/3 within Beloniformes, which I can confirm herein.

Intramandibular muscles possibly act in positioning the jaw (Karrer, 1967: “Stellbewegung”). Hertwig (2005) and Hertwig (2008) mentioned the reduction of intramandibular muscles and found a correlation between the loss of those muscles and a reduced mobility of particular bone elements. For Empetrichthys latos (Cyprinodontiformes), he noticed an ontogenetic reduction of intramandibular muscles. The movement of upper jaw bones in Beloniformes may be coupled to the movement of the lower jaw (see above) and hence they may underlie large mechanical stresses in fish hunting species. To withstand those forces, the bones of the lower jaw may have a higher degree of fusion resulting in the tendency to reduce intramandibular musculature.

Like Hertwig (2008), I defined an intramandibular portion of A2/3 as present in Pe. fluviatilis. However, the configuration of the intramandibular musculature of Pe. fluviatilis could be interpreted differently. In the present study, two intramandibular muscles were differentiated. First, an intramandibular portion of A2/3 is described as originating from the tendon of A2/3 by only a few muscle fibres. It narrowly inserts on the medial face of the anguloarticulare. Second, an intramandibular m. adductor mandibulae (Aω) is described, which is tendinously originating from the preopercular and the quadrate. That muscle has a flat insertion medially to the dentary, to cartilago Meckeli, and to the anguloarticular.

In contrast, Osse (1969) only described one intramandibular muscle for Pe. fluviatilis. That muscle, “Aω” in Osse (1969), has one origin at the tendon of A2/3. This “Aω” also has a narrow attachment to the anguloarticular, one tendinous attachment to the prearticular/quadrate and one flat insertion to the medial face of the lower jaw. Osse (1969) combined the Aω and the intramandibular portion of A2/3 of the present study as his “Aω.” Therefore, he did not differentiate the course of muscle fibres and other associated structures. The fibres of the intramandibular portion of A2/3 of the present study run anteroventrad. The fibres of the Aω were found to originate as a double fibred muscle from the tendon originating from the prearticular/quadrate. However, some fibres also originate from the tendon of A2/3, which is only partly fused with the tendon of Aω. While both tendons fuse, the course of the Aω-tendon is still separable (Fig. 5D). The fusion of the tendons and the origin of some Aω-fibres at the A2/3-tendon may have persuaded Osse (1969) to define only one intramandibular muscle.

One additional interpretation of intramandibular muscle configuration is possible. If a tendinous insertion of A2/3 to the tendon of Aω is hypothesised, the origin of some Aω-fibres may have been shifted to the tendon of A2/3. In that case, no intramandibular portion of A2/3 would exist in Pe. fluviatilis. If this configuration is a plesiomorphic condition of Acanthopterygii, the character should also be interpreted as a reversal within Beloniformes. In contrast, if one hypothesises the intramandibular portion of A2/3 to be independently reduced in Pe. fluviatilis, the character should be considered as homoplastic in Pe. fluviatilis (Percomorpha) and Beloniformes. To clarify that controversy, additional species of Percomorpha and Acanthopterygii need to be observed in great detail, but this was outside the scope of the present study.

Intramandibular section of the m. adductor mandibulae complex (Aω)

The intramandibular section of the m. adductor mandibulae complex (Aω) connects the suspensoric with the medial face of the lower jaw.

Origin. It originates with a tendon anteriorly at the medial face of the symplectic in Pa. brachypterus (Figs. 14 and 15) and D. pussila (Figs. 16 and 17) [state 0]. It originates directly at the ventral and the anterior edge of the quadrate in O. latipes (Figs. 12A–12C and 13) [state 1]. In R. corsula (Figs. 6 and 7), B. belone (Fig. 18), and S. saurus (Figs. 19 and 20), Aω originates broadly on the medial face of the quadrate and a part of the muscle can have a tendinous origin [state 2]. It attaches with a tendon anteroventrally to the medial face of the quadrate in At. boyeri (Figs. 8, 9 and 12D) and Ap. lineatus (Figs. 10 and 11) [state 3]; and in Pe. fluviatilis (Fig. 5), it originates with a tendon anteriorly at the medial face of the horizontal aspect of the preopercular and to a small amount medially at the middle area of processus caudalis quadrati [state 4]. The Aω is absent in X. oophorus [state 5].

Hertwig (2005) defined as a common character of hemiramphids: The origin of the flat tendon of Aω is situated at a part of the symplectic, which points rostrad. He studied species of Hyporhamphus, Nomorhamphus, and Hemiramphodon. Due to the diverging observation in D. pussila herein (Figs. 16 and 17), this character on the origin of Aω cannot be confirmed to be diagnostic for all hemiramphids. However, as that character was also found in Pa. brachypterus (Figs. 14 and 15), a potential synapomorphic character of (Exocoetidae + Hemiramphidae) is identified and a possible monophyly of Hemiramphidae could be indicated (Rosen, 1964; Rosen & Parenti, 1981; Collette et al., 1984). This would contradict the works of Lovejoy, Iranpour & Collette (2004) and Aschliman, Tibbetts & Collette (2005), who found “Hemiramphidae” paraphyletic. In the work of Lovejoy, Iranpour & Collette (2004), the Zenarchopteridae (among others Dermogenys, Hemiramphodon, Nomorhamphus) oppose the paraphyletic “Belonidae” (incl. Scomberesocidae) and Hyporhamphus belongs to a group, which opposes (Zenarchopteridae + “Belonidae”). Several species of “Hemiramphidae” that are closely related to Exocoetidae in the work of Lovejoy, Iranpour & Collette (2004), as well as several other species of the remaining groups of Beloniformes need to be observed to gain a better understanding on how that character is distributed. The absence of Aω was documented for some atherinomorph species by Hertwig (2008) and the reduction must have occurred several times independently.

Shape. In R. corsula (Figs. 6 and 7), Aω is separated into two heads at the level of the quadrate. The lateral head inserts broadly to the medial face of the dentary and cartilago Meckeli. The medial head of Aω inserts ventrally to the medial face of the dentary and anteriorly to the medial face of the anguloarticular [state 0]. The Aω represents a double-feathered muscle in Pe. fluviatilis (Fig. 5), At. boyeri (Figs. 8, 9 and 12D), D. pussila (Figs. 16 and 17), B. belone (Fig. 18), and S. saurus (Figs. 19 and 20), in which one of the muscle parts may project to a far caudad direction [state 1]. The Aω is a parallel fibred muscle in O. latipes (Figs. 12A–12C and 13) and Pa. brachypterus (Figs. 14 and 15) [state 2] and a simple feathered muscle in Ap. lineatus (Figs. 10 and 11) [state 3].

Insertion. On the medial face of the lower jaw, the Aω (when not differentiated into heads) inserts broadly to the dentary, cartilago Meckeli and/or to the anguloarticular in Pe. fluviatilis (Fig. 5), At. boyeri (Figs. 8, 9 and 12D), Pa. brachypterus (Figs. 14 and 15), and D. pussila (Figs. 16 and 17) [state 0]. It inserts broadly to the dentary, to the anguloarticular, and to the cartilago Meckeli, whereby a ventral part in feathered muscles inserts far anteriorly to the medial face of the dentary in B. belone (Fig. 18) and S. saurus (Figs. 19 and 20) [state 1]. It inserts to the ventral part of the dentary in Ap. lineatus (Figs. 10 and 11) [state 2] and posteriorly to the dentary and medially at the cartilago Meckeli in O. latipes (Figs. 12A–12C and 13) [state 3].

Hertwig (2005) and Hertwig (2008) has shown that the configuration of Aω is highly variable among Cyprinodontiformes. In comparison, this can also be concluded for the species observed herein.

M. intermandibularis

Cross section. M. intermandibularis connects the contralateral dentaries at their medial faces. The cross-section of m. intermandibularis is +/− round in At. boyeri (Figs. 8, 9 and 12D) [state 0]. It is big-bellied oval in Pe. fluviatilis (Fig. 5), R. corsula (Figs. 6 and 7), O. latipes (Figs. 12A–12C and 13), and X. oophorus (Fig. 3); i.e., it is at its maximum twice as broad as high [state 1]. It is elongated oval in Ap. lineatus (Figs. 10 and 11), Pa. brachypterus (Figs. 14 and 15), D. pussila (Figs. 16 and 17), B. belone (Fig. 18), and S. saurus (Figs. 19 and 20); i.e., it is (mostly much) more than twice as broad as high [state 2].

In each species studied, several specimens were observed and a tendency of a rounder cross-section of the muscle was found in At. boyeri (Figs. 8, 9 and 12D). In addition, the assignment to big-bellied or elongated oval has to be understood as a tendency in the variability of the specimens observed.

Shape. The m. intermandibularis is parallel fibred and has no tendinous origin at the dentary in Pe. fluviatilis (Fig. 5), At. boyeri (Figs. 8, 9 and 12D), Pa. brachypterus (Figs. 14 and 15), D. pussila (Figs. 16 and 17), B. belone (Fig. 18), and S. saurus (Figs. 19 and 20) [state 0]. However, it is spindle-shaped and has a tendinous origin at the dentary in R. corsula (Figs. 6 and 7), Ap. lineatus (Figs. 10 and 11), O. latipes (Figs. 12A–12C and 13), and X. oophorus (Fig. 3) [state 1].

M. protractor hyoidei

Origin. The m. protractor hyoidei connects the branchial apparatus with the lower jaw. It originates laterally at the ceratohyal in Pe. fluviatilis (Fig. 5), Pa. brachypterus (Figs. 14 and 15), and D. pussila (Figs. 16 and 17) [state 0], with two heads ventrally and laterally at the ceratohyal and at the anterior tips of the branchiostegal rays in O. latipes (Figs. 12A–12C and 13) [state 1], ventrally to the ceratohyal in R. corsula (Figs. 6 and 7), At. boyeri (Figs. 8, 9 and 12D), Ap. lineatus (Figs. 10 and 11), and X. oophorus (Fig. 3) [state 2] and medially to the ceratohyal in B. belone (Fig. 18) and S. saurus (Figs. 19 and 20) [state 3].

Course. A fusion with the contralateral m. protractor hyoidei occurs at the level of the jaws or suspensoric and united, they travel rostrad in R. corsula (Figs. 6 and 7), Ap. lineatus, O. latipes (Figs. 12A–12C and 13), X. oophorus (Fig. 3) (in relation to the jaw joint, the protractor fuses more anteriorly in X. oophorus when compared to O. latipes), Pa. brachypterus (Figs. 14 and 15), and D. pussila (Figs. 16 and 17) and anteroventrally at the fused mm. protractor hyoidei a tendon can be formed on each side [state 0]. Such a fusion does not occur in Pe. fluviatilis (Fig. 5), B. belone (Fig. 18), and S. saurus (Figs. 19 and 20) [state 1]. In At. boyeri (Figs. 8, 9 and 12D), at the level of the anguloarticular, the muscles fuse only in their ventral regions; they separate on the level of the dentary in order to insert independently of the contralateral muscle to the dentary [state 2].

Anterior part. When reaching m. intermandibularis, m. protractor hyoidei has a flat shape in Ap. lineatus (Figs. 10 and 11), O. latipes (Figs. 12A–12C and 13), and X. oophorus (Fig. 3) [state 0], or it is about as broad as high in Pe. fluviatilis (Fig. 5), At. boyeri (Figs. 8, 9 and 12D), Pa. brachypterus (Figs. 14 and 15), D. pussila (Figs. 16 and 17), B. belone (Fig. 18), and S. saurus (Figs. 19 and 20) [state 1]. At this level, m. protractor hyoidei already differentiated into two heads. The dorsal head is flat and the ventral head is as high as broad in R. corsula (Figs. 6 and 7) [state 2].

When reaching the dentary, the flat shape of the muscle in Ap. lineatus (Figs. 10 and 11) and O. latipes could be hypothesized as being an autapomorphic character of Cyprinodontoidei sensu Rosen (1964) (Fig. 1A).

Insertion. M. protractor hyoidei inserts dorsally to the insertion of m. intermandibularis at the dentary and covers at least the posterodorsal area of the latter muscle in Ap. lineatus (Figs. 10 and 11), Pa. brachypterus (Figs. 14 and 15), and S. saurus (Figs. 19 and 20) [state 0]. In Pe. fluviatilis (Fig. 5), R. corsula (Figs. 6 and 7), O. latipes (Figs. 12A–12C and 13), D. pussila (Figs. 16 and 17), and B. belone (Fig. 18), it inserts ventrally to the m. intermandibularis at the dentary [state 1]. It inserts dorsally as well as ventrally of m. intermandibularis to the dentary in X. oophorus (Fig. 3) [state 2].

Insertion tendon. The ventral part of m. protractor hyoidei extends into a long tendon, which reaches the anterior tip of the lower jaw in D. pussila (Figs. 16 and 17) and B. belone (Fig. 18) [state 0]. It does not extend into a long tendon to reach the anterior tip of the lower jaw in Pe. fluviatilis (Fig. 5), R. corsula (Figs. 6 and 7), At. boyeri (Figs. 8, 9 and 12D), O. latipes (Figs. 12A–12C and 13), and X. oophorus (Fig. 3) [state 1].

The anteroventral elongation of musculature in the region of the dentary seems to be associated with the elongated lower jaw within Beloniformes. In D. pussila (Figs. 16 and 17) and B. belone (Fig. 18), also a ventral insertion of m. adductor mandibulae (Aω) to the anterior tip of the lower jaw can be recognised. Besides the latter muscle, m. intermandibularis is also extended far rostrad in S. saurus (Figs. 19 and 20), however, in this species m. protractor hyoidei does not reach the anterior tip of the lower jaw. Referring to Haszprunar (1998), one could argue that the elongation of a muscle within the lower jaw is simply an adaptation correlated to food ingestion and hence would not have a value for phylogenetic questions. However, as noted by de Pinna (1991) and Haas (2003), such adaptations can be informative at particular hierarchical levels.

M. adductor arcus palatini

Origin and insertion. The anterior portion of m. adductor arcus palatini, the only portion of this muscle studied herein, originates along the whole parasphenoid and inserts dorsally along the entire suspensoric in R. corsula (Figs. 6 and 7), Ap. lineatus (Figs. 10 and 11), O. latipes (Figs. 2E, 12A–12C and 13), X. oophorus (Fig. 3), and D. pussila (Figs. 2G, 16 and 17) (in addition to other small attachments) [state 0]. In contrast, it originates on the posterior part of the parasphenoid and inserts on the posterior region of the suspensoric in Pe. fluviatilis (Figs. 2A and 5), At. boyeri (Figs. 2C, 8, 9 and 12D), Pa. brachypterus (Figs. 2F, 14 and 15), B. belone (Figs. 2H and 18), and S. saurus (Figs. 19 and 20) [state 1].

M. levator arcus palatini

M. levator arcus palatine originates on the skull roof behind the eye, runs ventrally, and inserts dorsally to the posterior part of the suspensoric.

Origin. It originates broadly on the sphenotic in Pa. brachypterus (Figs. 2F, 14 and 15) and D. pussila (Figs. 2G, 16 and 17) [state 0]. In R. corsula (Figs. 2B, 6 and 7), B. belone (Figs. 2H and 18), and S. saurus (Figs. 19 and 20), it originates on a ridge of the sphenotic, the processus sphenoticus, and some fibres originate directly on the sphenotic [state 1]. The muscle arises via a short tendon from the ventral edge of the transverse process of the sphenotic and runs ventrad along the posterior margin of the orbit, dorsally from the hyomandibular, and with few fibres from the sphenotic in O. latipes (Figs. 2E, Figs. 12A–12C and 13) and X. oophorus (Fig. 3) [state 2]. It originates ventrally at the dermosphenotic in At. boyeri (Figs. 2C, 8, 9 and 12D) [state 3] and from the autosphenotic and with some fibres at the sphenotic in Pe. fluviatilis (Figs. 2A and 5) and Ap. lineatus (Figs. 2D, 10 and 11) [state 4].

The m. levator arcus palatini plesiomorphically originates at the autosphenotic and with some fibres at the sphenotic. This condition is also visible in Ap. lineatus (Figs. 10 and 11) and could be assumed as being plesiomorphic for all Cyprinodontiformes (compare to Hertwig, 2005; Karrer, 1967).

Course. During its course from origin to insertion, the thickness of m. adductor arcus palatini hardly changes in R. corsula (Figs. 2B, 6 and 7), O. latipes (Figs. 2E, 12A–12C and 13), and X. oophorus (Fig. 3) [state 0], whereas in all other species it becomes more than twice as thick [state 1].

Relation to other muscles. M. levator arcus palatini runs dorsally of the medial and lateral head of A2/3 and does not run between both heads heads in Pe. fluviatilis (Figs. 2A and 5), R. corsula (Figs. 2B, 6 and 7), Ap. lineatus (Figs. 10 and 11), O. latipes (Figs. 2E, 12A–12C and 13), X. oophorus (Fig. 3), and Pa. brachypterus (Figs. 2F, 14 and 15) [state 0]. It is clearly situated between the lateral and the medial head of A2/3 in D. pussila (Figs. 16 and 17) [state 1] or it is only partly surrounded by the lateral and by the medial head of A2/3 in B. belone (Figs. 2H and 18) and S. saurus (Figs. 19 and 20) [state 2].

Insertion. On the lateral face of the suspensoric of Pe. fluviatilis (Fig. 5), Ap. lineatus (Figs. 10 and 11), Pa. brachypterus (Figs. 14 and 15), D. pussila (Figs. 16 and 17), B. belone (Fig. 18), and S. saurus (Figs. 19 and 20), m. levator arcus palatini inserts onto the hyomandibular and to the metapterygoid and with some fibres, it also can attach anteriorly to the processus lateralis hyomandibularis [state 0]. In O. latipes (Figs. 12A–12C and 13) and X. oophorus (Fig. 3), it inserts on the broad face of the praeopercular and posterodorsally to the symplectic [state 1]. In R. corsula (Figs. 6 and 7) and At. boyeri (Figs. 8, 9 and 12D), it inserts on the hyomandibular, anteriorly to the processus lateralis hyomandibularis, to the metapterygoid, and to the broad face of the preopercular [state 2].

Kulkarni (1948) identified the metapterygoid as being reduced within Adrianichthyidae. This suggestion was only based on his observations in Horaichthys setnai and O. melastigma. Werneburg & Hertwig (2009) identified a horizontal suture in the ‘symplectic’ (sensu Kulkarni, 1948) of O. latipes, which could represent the border of the metapterygoid. In histological sections and hence in 3D reconstructions (Werneburg & Hertwig, 2009), such a differentiation of bones was not visible. As such, the situation remains unclear.

M. dilatator operculi

Origin. M. dilatator operculi connects the opercle with the skull roof. It originates ventrally at the lateral face of the sphenotic in Pa. brachypterus (Figs. 2F, 14 and 15), X. oophorus (Fig. 3), and S. saurus (Figs. 19 and 20) [state 0]. It originates laterally at the sphenotic, at the autosphenotic, and with some fibres possibly at the anteroventral area of the pterotic in Pe. fluviatilis (Figs. 2A and 5), At. boyeri (Figs. 2C, 8, 9 and 12D), and Ap. lineatus (Figs. 2D, 10 and 11) [state 1]. In R. corsula (Figs. 2A, 6 and 7), O. latipes (Figs. 2E, 12A–12C and 13), D. pussila (Figs. 2G, 16 and 17), and B. belone (Figs. 2H and 18), it originates laterally at the sphenotic and anteriorly at the lateral face of the pterotic [state 2].

Shape. Anteriorly, m. dilatator operculi extends almost to the eye and lies dorsally to m. levator arcus palatini in R. corsula (Figs. 2A, 6 and 7), At. boyeri (Figs. 2C, 8, 9 and 12D), Ap. lineatus (Figs. 2D, 10 and 11), and O. latipes (Figs. 2E, 12A–12C and 13) [state 0]. It does not reach the eye region in Pe. fluviatilis (Figs. 2A and 5), X. oophorus (Fig. 3), Pa. brachypterus (Figs. 2F, 14 and 15), D. pussila (Figs. 2G, 16 and 17), B. belone (Figs. 2H and 18), and S. saurus (Figs. 19 and 20) [state 1].

M. levator operculi

Origin. The m. levator operculi connects the opercle with the skull roof. It is an undivided muscle with an origin ventrally at the lateral face of the pterotic in all taxa studied [state 0], except for Pe. fluviatilis. In this species is a bipartite muscle with a large anterior origin ventrally at the lateral face of the pterotic and a small posterior origin ventrally at the ventral situated extrascapula (Figs. 2A and 5) [state 1].

Insertion. M. levator operculi inserts dorsally to the medial face of the opercle and has a continuous horizontal level of insertion in Pe. fluviatilis (Fig. 5), R. corsula (Figs. 6 and 7), Ap. lineatus (Figs. 10 and 11), O. latipes (Figs. 12A–12C and 13), X. oophorus (Fig. 3), Pa. brachypterus (Figs. 14 and 15), and D. pussila (Figs. 16 and 17) [state 0]. It also inserts dorsally at the medial face of the opercle in B. belone (Fig. 18) and S. saurus (Figs. 19 and 20), but it attaches more ventrally to the anterior region of the medial face of the opercle [state 1]. The muscle inserts dorsally to the medial face and dorsally to the lateral face of the opercle in At. boyeri (Figs. 8, 9 and 12D) [state 2].

Nerves

Truncus maxillaris infraorbitalis trigemini. The truncus maxillaris infraorbitalis trigemini branches into the ramus mandibularis trigemini and ramus maxillaris trigemini short before or after leaving the neurocranium in Pe. fluviatilis (Fig. 5), R. corsula (Figs. 6 and 7), At. boyeri (Figs. 8, 9 and 12D), and Ap. lineatus (Figs. 10 and 11)—and dorsally to the suspensoric, the ramus mandibularis trigemini covers the ramus maxillaris trigemini laterally [state 0]. Contrary, in O. latipes (Figs. 12A–12C and 13), X. oophorus (Fig. 3), Pa. brachypterus (Figs. 14 and 15), and D. pussila (Figs. 16 and 17), it first branches at the level of the eye [state 1]. In B. belone (Fig. 18) and S. saurus (Figs. 19 and 20), it branches already within the neurocranium. Afterwards, the ramus maxillaris trigemini splits into two branches. Dorsally to the posterior part of the suspensoric, the branches align laterally and medially along the course of ramus mandibularis trigemini. On the level of the jaw joint, the branches of ramus maxillaris trigemini change their course into an anterodorsad direction and enter the upper jaw. Ramus mandibularis trigemini travels anteroventrad to the lower jaw [state 2].

Ramus mandibularis facialis. The ramus mandibularis facialis branches after leaving the hyomandibular laterally to the suspensoric in order to run with two branches to the medial side of the suspensoric in At. boyeri (Figs. 8, 9 and 12D), Ap. lineatus (Figs. 10 and 11), B. belone (Fig. 18), and S. saurus (Figs. 19 and 20) [state 0]. In Pe. fluviatilis (Fig. 5), R. corsula (Figs. 6 and 7), O. latipes (Figs. 12A–12C and 13), Pa. brachypterus (Figs. 14 and 15), and D. pussila (Figs. 16 and 17) it branches differently [state 1]. The course of that nerve could not be followed in X. oophorus (Fig. 3).

Ligaments

Lig. premaxillo-maxilla. This ligaments spans broadly between premaxilla and maxilla in B. belone (Fig. 18) and S. saurus (Figs. 19 and 20) [state 0] and between the proximal ends of the premaxilla and the maxilla in all other species [state 1].

Hertwig (2008) argued for the absence of the ligament in Beloniformes and mentioned an extensive area of connective tissue instead. Based on arguments of Werneburg (2013b), I homologise this tissue with the broad ligament found in other taxa.

Primordial ligament. This ligament is present as a lig. maxillo-anguloarticulare between the maxilla and the anguloarticular in Pe. fluviatilis (Figs. 2A and 5) and At. boyeri (Figs. 2C, 8, 9 and 12D) [state 0]. The ligament is absent in all other species [state 1].

Upper jaw/palatine ligament. A ligament, which connects the palatine and the upper jaw, is present as lig. palato-maxilla between palatine and maxilla in At. boyeri (Figs. 8, 9 and 12D), Ap. lineatus (Figs. 10 and 11), O. latipes (Figs. 12A–12C and 13), X. oophorus (Fig. 3), and Pa. brachypterus (Figs. 14 and 15) [state 0]. It is present as lig. palato-premaxilla between palatine and premaxilla in Pe. fluviatilis (Fig. 5) [state 1] or is absent in R. corsula (Figs. 6 and 7), D. pussila (Figs. 16 and 17), B. belone (Fig. 18), and S. saurus (Figs. 19 and 20) [state 2].

An autapomorphy in the ground pattern of Atherinomorpha may be the presence of a lig. palato-maxilla. The absence of the ligament in R. corsula (Figs. 6 and 7) and a different attachment of the ligament in Pe. fluviatilis makes it impossible to reconstruct the ground pattern.

Lig. parasphenoido-suspensorium. This ligament is present in Pe. fluviatilis (Fig. 5), At. boyeri (Figs. 8, 9 and 12D), and S. saurus (Figs. 19 and 20) [state 0]. It is absent in all other species [state 1].

For Pe. fluviatilis, Osse (1969) described two ligaments (his No. XVII and XVIII) that originate from the parasphenoid and insert to the dorsal edge of the suspensoric. This differentiation of the ligament could not be identified in the manual dissections performed for the present study.

Conclusions

In the present study, the variety of jaw, suspensoric, and opercle muscles was described for several acanthopterygian fishes with a focus on Beloniformes. The diversity of jaw muscles within Beloniformes corresponds to the external differences in their jaw morphology. As such, long beaked forms and species with protractible mouths show remarkable differences in their jaw musculature that may be correlated to stiffening or high mobility of the jaws.

Most important anatomical differences detected in this study exist in the external jaw musculature of Beloniformes. The jaw adductors belong to the most intensely studied muscles in vertebrates due to their prominent size and variation in the head and their importance for feeding mechanisms (Haas, 2001; Diogo, 2008; Diogo & Abdala, 2010; Daza et al., 2011; Konstantinidis & Harris, 2011; Werneburg, 2013a; Werneburg, 2013b; Datovo & Vari, 2013; Datovo & Vari, 2014). Among Acanthopterygii, the external section of m. adductor mandibulae (A1) experienced comprehensive diversifications (Wu & Shen, 2004), and among Beloniformes, it can either be present or absent.

The A1 lowers the upper jaw in most fishes. As an autapomorphy of Beloniformes, Mickoleit (2004) mentioned the reduced mobility of bones related to the upper jaw. Hertwig (2005) hypothesised that the reduced mobility of those bones might be correlated with the reduction of A1 within Beloniformes or the displacement of the A1-insertion apart from the upper jaw. In the present study, such a replacement of A1 was discovered in O. latipes (Fig. 2E; see also Werneburg & Hertwig, 2009). This species can still move its upper jaw during feeding (I Werneburg, pers. obs., 2006), which questions the possibility of a functional correlation of the character pair mentioned by Hertwig (2005) and Hertwig (2008), namely ‘A1 no longer attached to upper jaw’ and ‘non-moveable upper jaw bones’.

Moreover, in the flying fish Pa. brachypterus, which has no A1 (Fig. 2F), a protrusible jaw was discovered herein. Therefore, the upper jaw bones are moveable against each other (Figs. 14 and 15).

The hemiramphid Dermogenys pusilla, which hunts at the surface of the water (Meisner, 2001), is able to easily move its short upper jaw, although the species has no A1 (Fig. 2G). Hence, coupled by ligament attachments, the lifting of the upper jaw appears to be indirectly performed by lowering the lower jaw. A deep coupling of those structures can be hypothesised for most other A1-lacking Beloniformes. In addition, the mobility of the protrusible upper jaw of Pa. brachypterus suggests a strong ligament-bone interaction (Figs. 14 and 15).

Among hemiramphids, whose phylogenetic relationship is debated, A1 can be absent (this study: Dermogenys pussila; Hertwig, 2008: Hyporhamphus unifasciatus) or can be present (Hertwig, 2008: Nomorhamphus sp., Hemiramphodon phaiosoma; Rosen, 1964: Arrhamphus brevis). Also Exocoetidae seem to have members with an A1 (Wu & Shen, 2004: Cypselurus cyanopterus, Parexocoetus mento; but see comments in the Results section) and members without an A1 (this study: Pa. brachypterus). The phylogenetic significance of those conditions can first be adequately estimated when more species are observed and more clarity exists about phylogenetic interrelationship. But this requires further detailed and comprehensive observations.

At least for B. belone (Fig. 2H) and S. saurus (Figs. 19 and 20), one may hypothesise that the loss of the A1 could be related to a strong fixation of the upper jaw to the cranium, realised by lig. premaxillo-frontale. Whether the upper jaw of both species is still moveable in vivo is not known so far, but is not expected.

As seen in hemiramphids, an elongated lower jaw not necessarily involves the reduction of A1. Xenopoecilus oophoris, an adrianichthyid with duckbill-like jaws, also has an A1 (Fig. 3), which is attached to the upper jaw. This indicates that also an elongated upper jaw, which possibly was present in the ground pattern of Beloniformes already (Parenti, 1987), not necessarily implies the loss of A1. Only the derived condition of two species, B. belone and S. saurus, which possess a stiffened upper jaw, may be clearly correlated to the loss of A1. As such, it can be expected that another belonid, Potamorrhamphis eigenmannii (Miranda Ribeiro, 1915), which has a moveable upper jaw in vivo (I Werneburg, pers. obs., 2006), could have an A1, but this hypothesis needs further observation. The present study shows that the loss of A1 must not be interpreted only in correlation to elongated jaws. Other biomechanical requirements must be considered.

The studied selection of non-beloniform species must be handled with care when choosing them as potential outgroup species (as example see Hertwig, 2008). Compared to the insufficient documentation of the cranial musculature of most acathopterygian groups, the species dissected herein appear to show several derived characters. E.g., Rh. corsula has three main components of A2/3. Most mugiliform taxa, however, are reported to have a different arrangement of that muscle (Gosline, 1993: Agonostomus; Van Dobben, 1935: Mugil; Wu & Shen, 2004: Chelon, Crenimugil; Starks, 1916: Mugil; Eaton, 1935: Mugil). As the authors of these studies did not observe histological sections, these findings could represent artefacts caused by the lower resolution of manual dissection.

As representative of the potential sister group to all remaining Beloniformes, the adrianichthyids Oryzias latipes and Xenopoecilus oophorus were studied herein. Hertwig (2005), Hertwig (2008) and Werneburg & Hertwig (2009) already diagnosed several derived characters for O. latipes that could be affirmed herein and together with X. oophorus, it shares several derived characters. Due to the distinctive morphology of Adrianichthyidae, problems could arise when reconstructing the jaw muscle configuration in the ground pattern of Beloniformes. In addition to several derived characters, the taxon seems to display several plesiomorphic characters shared with Cyprinodontiformes. This finding persuaded Rosen (1964) and Li (2001) to postulate a sister group relationship of Adrianichthyidae + Cyprinodontiformes, named as Cyprinodontoidei (Fig. 1A). The present study highlights which characters are most variable among near related species and may assist taxon and character selection in future phylogenetic studies.

The differing external jaw morphology of diverse beloniform fishes is nicely reflected in the anatomy of their jaw musculature. Apparent changes concern the absence or presence of the A1 and arrangements of the intramandibular musculature. Both muscles are coupled to the upper or lower jaw, which are connected by ligaments themselves. The strong attachment of the upper jaw to the neurocranium, as visible in needlefishes and sauries, involves complex rearrangements of the soft tissue of the jaw apparatus.

I am grateful to Stefan T. Hertwig for discussion and advice (but I am of course responsible for any remaining mistakes). I thank Lynne Parenti, Rolf Beutel, Stefan T. Hertwig, and Manfred Schartl for kindly providing specimens. Janine M. Ziermann, Martin Fischer, Lennart Olsson, and Torsten M. Scheyer are thanked for discussion. Rommy Peterson and Katja Felbel helped with laboratory concerns. Julio Mario Hoyos, Janine M. Ziermann, Laura A.B. Wilson, and anonymous reviewers gave constructive critics to former versions of the manuscript. I am most grateful to Marcelo R. Sánchez-Villagra and Walter G. Joyce to for their generous support of my research.

Additional Information and Declarations

Competing Interests

Author Contributions

Animal Ethics

The author declares there are no competing interests.

Ingmar Werneburg conceived and designed the experiments, performed the experiments, analyzed the data, wrote the paper, prepared figures and/or tables.

The following information was supplied relating to ethical approvals (i.e., approving body and any reference numbers):

No permission was required. The vertebrate animals used in this study were manually dissected. The specimens were ethanol preserved and stem from scientific collections as indicated in the methods section of the manuscript.

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
