# Peer review of "Morphology of the jaw, suspensorial, and opercle musculature of Beloniformes and related species (Teleostei: Acanthopterygii), with a special reference to the m. adductor mandibulae complex"

_PeerJ, doi:10.7717/peerj.769_

## Round 0.1 · original submission · Minor Revisions

· Academic Editor

Minor Revisions

Two reviewers have evaluated the ms and provided their comments below, I also have some additional small comments to the paper, below.

Overall, both reviewers think this paper is excellent and will make an important contribution to the literature. I agree with their comments, and find the ms to be highly detailed and beautifully illustrated. Both reviewers have made some minor suggestions that will help the reader, and I ask the author to pay careful attention to those helpful and insightful comments. In particular, both reviewers have suggested some potential references to review and consider adding to the ms, reviewer #1 has suggested to add some additional introduction about the m.adductor mandibulae complex and consider changing the section headings in the paper, and reviewer #2 has suggested that a summary table will be helpful to the reader to rapidly gain an understanding on the potential synapomorphies, plesiomorphies and autapomorphies identified in this work.

Minor editorial comments:
Ln76: remove, “to”
Ln80: should be “studied in very few species”
Ln82: suggest changing this line to “insufficiently illustrated, making broad phylogenetic comparisons impossible.”
Ln96: reference here?
Ln244: plesiomorphic
Ln251: suggest to rephrase to, “”their position of origin (or insertion) and spatial orientation were considered.”
Ln272: should be “difficult”
Ln404: better to write “outside” than “out of”
Ln441: add period after Insertion
Ln520: “at” rather than “on” ?
Ln549: suggest rephrasing to, “and X. oophorus (Fig. 3), whereas in all other species it becomes more than twice as thick.”
Ln561: insert “the” before praeopercular
Ln643: suggest “manual dissections performed for the present study”
Ln653: high mobility of “the” jaws

·

Basic reporting

Dear Editor,

The manuscript entitled: "Morphology of the suspensorial, jaw, and opercle musculature of Beloniformes and related species (Teleostei: Acanthopterygii), with a special reference to the m. adductor mandibulae complex" is a great example of detailed anatomical description.
The author focuses on the description and discussion of jaw, opercle and suspensorial musculature and discusses his finding in Beloniformes with respect to potential phylogenetic relevant characters. The description of the muscles, ligaments and nerves chosen shows an outstanding quality of detail accompanied by high quality visual documentation (slides, pictures).

However, there are a couple of minor points that should be addressed before a final publication:

General comments:

Headline: The title should be changed in the sense that the jaw is more anterior than the suspensorium and opercle and also the majority of muscles and ligaments that are described are related to the jaw. Therefore, the headline should be “Morphology of the jaw, …”

Introduction: The author emphasis in the headline the m. adductor mandibulae complex and the major part of his results is about muscles of this complex. However, it would be helpful to provide a brief introduction to this complex. There are over 1000 articles referring to this muscle complex in teleost. It should not be the aim to review all of them, but a sentence to the diversity and problematic of this muscle complex should be provided – a Review that could be added to that (for readers that would like to go into more detail): Datovo, A., & Vari, R. P. (2013). The jaw adductor muscle complex in teleostean fishes: Evolution, homologies and revised nomenclature (Osteichthyes: Actinopterygii). PloS one, 8(4), e60846.

Results: The result part is a mix of very detailed and accurate description and discussion of the observations in comparison to former studies on Beloniformes and related species. As those discussions in the result part seem to be a very crucial for the phylogenetic interpretation of one character, I suggest to rename the Results into “Results and Discussion”. The actual Discussion should be renamed Conclusions as it summarizes with little additional information the Result part. The actual Conclusion headline should be then be removed.

Specific comments:
line 60: “in contrast to several analyses” – Which analyses? Specify: add citations or indicate if those analyses are molecular, morphological …
line 62-64: sort the citation in chronological order as done throughout the rest of the manuscript
line 76: delete “to”
line 158: ancestrally – of what / to what? Also, this sentence might be confusing as the adductor mandibulae is already devided in chondrichthyes (e.g. Soares, M. C., & de Carvalho, M. R. (2013). Comparative myology of the mandibular and hyoid arches of sharks of the order hexanchiformes and their bearing on its monophyly and phylogenetic relationships (Chondrichthyes: Elasmobranchii). Journal of morphology, 274(2), 203-214.)
line 170: “First” – I couldn’t find “second”
line 260: “at its origin”
line 441: Add dot after Insertion.
line 504: Add “the” before m. intermandibularis
line 558: onto the hyomandibular
line 587: delete “also”
line 656: As the add.mand. complex is one of the most intensively studied muscles in vertebrates having only one publication (self-citation) is a bit confusing, as the citation is mainly about turtles. Just to name a few others:
Konstantinidis, P., & Harris, M. P. (2011). Same but different: ontogeny and evolution of the musculus adductor mandibulae in the Tetraodontiformes. Journal of Experimental Zoology Part B: Molecular and Developmental Evolution, 316(1), 10-20.
Datovo, A., & Vari, R. P. (2014). The adductor mandibulae muscle complex in lower teleostean fishes (Osteichthyes: Actinopterygii): comparative anatomy, synonymy, and phylogenetic implications. Zoological Journal of the Linnean Society, 171(3), 554-622.
Datovo, A., & Vari, R. P. (2013). The jaw adductor muscle complex in teleostean fishes: Evolution, homologies and revised nomenclature (Osteichthyes: Actinopterygii). PloS one, 8(4), e60846.
Daza, J. D., Diogo, R., Johnston, P., & Abdala, V. (2011). Jaw adductor muscles across lepidosaurs: a reappraisal. The Anatomical Record, 294(10), 1765-1782.
Haas, A. (2001). Mandibular arch musculature of anuran tadpoles, with comments on homologies of amphibian jaw muscles. Journal of Morphology,247(1), 1-33.

Experimental design

No Comments

Validity of the findings

This manuscript provides a great basis for further comparative studies, for both functional morphology and phylogeny.

·

Basic reporting

No comments.

Experimental design

No comments.

Validity of the findings

The conclusions must be the product of discussion, not a repetition of what was said there or in the results section. I think it should be more blunt, stating the most important and innovative findings of work.

Additional comments

COMMENTS ABOUT WERNEBURG PAPER
I think that the paper is very useful because there are not many descriptions about muscles in fishes, mainly head muscles descriptions. It is clear that these descriptions will be very important materials to resolve problems in biomechanics and systematic. I found that the paper is written very good , and it is very clear.
Results:
• I saw that the author did comparisons in results section; I am not sure if the author instructions accept this, but I would prefer that, in results, includes just descriptions, and comparisons with other descriptions be done in Discussion section.
• I think that it will be important to construct a table, as a summary, to remark the presence of potential synapomorphies, plesiomorphies and autapomorphies, although these ones are in the paper. I am sure that this table will be very useful for future studies in fishes.
Conclusions:

Other comments:
• 835 Parenti LR. 1987. Phylogenetic aspects of tooth and jaw structure of the Madeka Oryzias
836 latipes, and other beloniform fishes. JOURNAL OF ZOOLOGY 211:561-572. In lower case and upper case.
• Alexander RM. 1967. Mechanisms of the jaws of some atheriniform fish. JOURNAL OF
744 ZOOLOGY 151:233-255. In lower case and upper case.

• 794 Hertwig ST. 2008. Phylogeny of the Cyprinodontiformes (Teleostei, Atherinomorpha): the
795 contribution of cranial soft tissue characters. Zoologica Scipta 37:141-174: Change to Scripta
• 833 Osse JWM. 1969. Funktional morphology of the head of the perch (Perca fluvialitis L.):
834 An electromyographic study: Change to “functional”.
• Candewalle et al. (2002): It is not in the bibliography.
• 163 tendinous insertion to the upper or lower jaw (i.e. Allis, 1897). An A1 is present in Perca
164 fluviatilis (Fig. 3A, S1): Change to 2A.
• 194 Starks (1916): It is not in the bibliography.
In Pe. fluviatilis, the muscle is situated dorsolateral to the internal
• 202 section and the complete lateral head (A2/3, lateral) is not covered in lateral view (Fig. 3A, 203 S1): Change to 2A.

• 210 an outgroup of Atherinomorpha, the author used Pe. fluviatilis, in which the A2/3-portions
211 are situated above each other in a horizontal plane (Fig. 3A, S1): Change to 2A.

• Insertion. The tendon of A1 inserts on the lateral face of the anterior part of the maxilla
225 in Pe. fluviatilis (Fig. 3A, S1): Change to 2A.

• 259 Origin. In Pe. fluviatilis (Fig. 3A, S1) Change to 2A.

• 269 Jourdain (1878: It is not in the bibliography.

• 625 an extensive area of connective tissue instead. Based on arguments of Werneburg (2013b),: This is not in the bibliography.

• 699 eigenmannii (Miranda Ribeiro, 1915),: This is not in the bibliography.

• 708 a different arrangement of that muscle (Gosline, 1993: This is not in the bibliography.

• Van Dobben, 1935: It is not in the bibliography.

• Starks, 1916:It is not in the bibliography.

• Eaton, 1935: It is not in the bibliography.

I think that it will be useful that the author reads some chapters in these books, because they can be important for comparisons:
DIOGO, RUI. 2008. The Origin of Higher Clades: Osteology, Myology, Phylogeny and Evolution of Bony Fishes and the Rise of Tetrapods.
DIOGO, R. & ABDALA, V. 2010. Muscles of Vertebrates: Comparative Anatomy, Evolution, Homologies and Development

---

## Round 0.2 · accepted · Accept

· Academic Editor

Accept

Thank you very much for carefully modifying your manuscript in line with the earlier comments made by both reviewers. I am happy to see that you have responded to each comment, and have adopted all of their suggested minor modifications. I think the revisions have now helped the manuscript to be more accessible to the reader, and particularly the additional tables and figure are most useful in that regard.

Beyond a couple of very minor wording points, listed below, that you may wish to correct at the proof stage, I find no further revisions are required and I look forward to seeing your paper published.

Minor points:

Ln134: “data….were modified”
Ln188: I would suggest to delete, “orientation” and simply write, “They should serve as a summary of..”
Ln373: perhaps insert “studied here” after “species” ?
Ln1033” “of the literature” should be “from the literature”